

# Precipitation at Dumont d'Urville, Adélie Land, East Antarctica: the APRES3 dataset

Christophe Genthon[1], Alexis Berne[2], Jacopo Grazioli[3], Claudio Durán Alarcón[1], Christophe Praz[2], Brice Boudevillain[1]

[1]Univ. Grenoble Alpes, CNRS, IRD, Grenoble INP, IGE, F-38000 Grenoble, France
[2]Environmental Remote Sensing Laboratory, Environmental Engineering Institute, School of Architecture, Civil and Environmental Engineering, École Polytechnique Fédérale de Lausanne, CH-1015 Lausanne, Switzerland
[3]Federal Office of Meteorology and Climatology, MeteoSwiss, Locarno-Monti, Switzerland

*Correspondence to*: Christophe Genthon (christophe.genthon@cnrs.fr)

**Abstract:** Compared to the other continents and lands, Antarctica suffers a severe shortage of in-situ observations of precipitation. APRES3 (Antarctic Precipitation, Remote Sensing from Surface and Space) is a program dedicated to improve the observation of the Antarctic precipitation, both from the surface and from space, to assess climatologies and evaluate and ameliorate meteorological and climate models. A field measurement campaign was deployed at Dumont d'Urville station at the coast of Adélie Land in Antarctica, with an intensive observation period from November 2015 to February 2016 followed by continuous radar monitoring through 2016 and beyond. Among other results, the observations show that a significant fraction of precipitation sublimates in a dry surface katabatic layer before it reaches and accumulates at the surface, a result evidenced thanks to the profiling capabilities of precipitation radars. While the bulk of the data analyses and scientific results are published in specialized journals, this paper provides a compact description of the dataset now archived on PANGAEA data repository (https://www.pangaea.de, DOI: 10.1594/PANGAEA.883562) and made open to the scientific community to further its exploitation for antarctic meteorology and climate research purposes.

## 1 Introduction

The Antarctic ice sheet is a huge continental storage of water which, if altered through climate change, has potential to significantly affect global sea-level. While climate models consistently predict an increase in precipitation in the future in Antarctica (e.g. Palerme et al. [2016]), most of which falls in the form of snow that will not melt and thus will accumulate further ice, observational data to verify the current precipitation in the models are still in demand. Antarctica is the poor cousin of global precipitation observation and climatology building efforts. Citing Schneider et al. [2014] of the Global Precipitation Climatology Center (GPCC): "The GPCC refrains from providing a (precipitation) analysis over Antarctica" because of poor data coverage. GPCC's global maps of continental precipitation from in situ observations are consequently left blank over Antarctica. Satellites offer rising prospects to monitor remote, difficult and/or uninhabited regions, but even



then Antarctica tends to be excluded from comprehensive and/or global studies (e.g. Funk et al. [2014]). Only those studies that specifically focus on the polar regions and Antarctica have presented and discussed aspects of the Antarctic precipitation by satellite (Palerme et al. [2014], Behrangi et al. [2016], Palerme et al. [2016], Palerme et al. [2017]). Yet, in situ observations are still lacking to suitably calibrate and validate the satellite products.


The measurement of solid precipitation is notoriously difficult (Goodison et al. [1998], Nilu [2013]). Difficulties are exacerbated in Antarctica because access and operations are logistically difficult and environmental conditions are extreme. Antarctica is the driest continent on Earth in terms of precipitation: satellite data estimate the mean precipitation at 171 mm/yr of water north of 81°S, the latitude reached by the polar orbiting satellites [Palerme et al., 2014]. Low precipitation is

supported by net accumulation measurements at the surface using glaciological methods [Eisen et al., 2007] which yield equally low numbers [Arthern et al., 2006]. On the high Antarctic plateau, the accumulation is only a few cm/yr annually (e.g. Genthon et al. [2015]). Such low precipitation rate would be very hard to monitor even in more genial environments.  It can simply not be done with conventional instruments in Antarctica. Satellite data and glaciological reconstructions, as well as models and meteorological analyses, support a dry interior but indicate that precipitation is much larger at the peripheries

of the Antarctic ice sheet, yearly reaching several tens of cm, or even meters locally [Palerme et al. 2014]. However, there, strong katabatic winds frequently blow which adversely affect the conventional precipitation measurement methods. Collecting instruments (bucket-style instruments that capture and collect to measure snowfall, typically by weighing or tipping buckets counts) actually undercatch or overcatch because of air deflection and turbulence caused by the instruments themselves. In addition, they not only catch fresh falling snow but also drifting / blowing snow which was previously

deposited at the surface, then eroded and re mobilized by the strong winds. Non-catching instruments, including in-situ (disdrometer) and remote (radar, lidar) sensing instruments offer interesting prospects. Radars are particularly attractive because they can profile through the air layers. They can sense both horizontally to expand the spatial significance of the measurement, and vertically to scan the origin and fate of precipitation since condensation in the atmospheric column, from the clouds (see Witze [2016] for and An application in Antarctica) and above to the surface, and separate blowing snow in

the lower layers from precipitation higher up.

However, while radars are customarily used in other regions to monitor liquid precipitation (e.g. Krajewski and Smith [2002], Fabry [2015]), and many campaigns have also been conducted in high latitude and high altitude regions to study snowfall (e.g. Schneebeli et al. [2013], Grazioli et al. [2015], Medina and Houze [2015], Moisseev et al. [2015], Kneifel et al [2015]), experience is still limited in the Antarctic environment [Gorodetskaya et al., 2015]. Because such instruments do not

collect and directly measure the mass of falling precipitation, but rather measure the fraction of an emitted radiation which is reflected back by the hydrometeors, quantification in terms of precipitation involves both physically-based (electromagnetic laws of diffusion, diffraction and propagation) and hypothesis-based (particle population size and shape, habits) post



processing. The hypothesis-based part requires calibration and validation using various sources of in situ measurements (e.g. Souverijns et al. [2017]).

As part of the APRES3 project (Antarctic Precipitation, Remote Sensing from Surface and Space, http://apres3.osug.fr), starting in November 2015 until February 2016 for the intense observing period but still ongoing for some observations, an unprecedentedly comprehensive field campaign was launched at the French Dumont d'Urville Antarctic scientific station at

the coast of Adélie Land. The objective was to measure and monitor precipitation not only in terms of quantity but also of falling snow particle characteristics and microphysics. The range of instruments included a profiling K-band and a polarimetric scanning X-band radar, a multi-angle snowflake camera (MASC), an OTT Pluvio2 weighing gauge, and a Biral VPF-730 disdrometer. A weather station reporting temperature, moisture and wind conditions near the instruments was also deployed. Finally, a depolarization lidar was tentatively operated but run into problems and is not further mentioned here. All

instruments were removed at the end of January 2016 except the K-band radar which has remained in operation throughout 2016 and beyond. Grazioli et al. [2017a] provides a comprehensive description of the data and analysis techniques and discusses scientific outcomes. Further work is ongoing to address the calibration, verification and validation of meteorological and climate models and of satellite remote sensing techniques with the data. Meanwhile, because this is a unique dataset, dissemination to the wider community for similar use with other models and remote sensing processing

approaches or other research purposes is considered timely. This paper provides a compact description of the dataset and dissemination.

## 2 Dataset description

Grazioli et al. [2017a] provides ample information on the observation site, most instruments and methods. A summary and complementary informations are provide below.

### 2.1 Site description

The main APRES3 (austral) summer field campaign took place at the French Antarctic scientific station Dumont d'Urville (DDU) in Adélie Land (66.6628°S, 140.0014°E). The station is on Petrels Island located only ~5 km off the continent and the ice sheet proper: the observations are thus representative of the very coast of the Antarctic ice sheet. Because the station was operated for more than 60 years uninterruptedly, the means and statistics of meteorology and climate are documented

(König-Langlo et al. [1998], Grazioli et al. [2017a]). A main meteorological feature is the strong katabatic winds that frequently blow in the area. Adélie Land has been coined "the home of the blizzard" by Mawson [1915] after the 1[st] Australian Antarctic winter over in this region. However, much of the coasts of Antarctica are affected by the katabatic winds [Parish and Bromwich, 1987]. DDU is a perfect place to sample their consequences including in relation with precipitation.



## 2.2 Instruments and data

Standard measurements of atmospheric variables (temperature, wind speed, wind direction, relative and specific humidity, atmospheric pressure) are collected regularly all year long by the French meteorological service (Météo France), and a radiosounding is made daily at 00UTC. The routine program does not involve any instrumental measurement of precipitation. There are reports of visual estimation of the occurrence and type in METAR convention but no quantification. For the APRES3 campaign, several instruments were deployed from the beginning of November 2015 to the end of January

2016 to objectively characterize and quantify the occurrences and amounts of precipitation, as described below.

### 2.2.1 Surface-based remote sensing instruments

As reported in the introduction, traditional collecting precipitation gauges are unreliable in Antarctica in general, and in particular in the coastal regions strongly affected by katabatic winds. Radars are masterpieces of the APRES3 campaign. Radars remotely sense the hydrometeors, estimate quantities and speed and from this derive precipitation rates. Radars can

scan and profile through atmospheric and hydrometeor layers and look beyond blowing snow near the surface. Two radars were deployed: a K-band frequency-modulated continuous-wave profiler and an X-band dual polarization scanning Doppler radar. The 1$^{st}$ instrument, a Metek micro-rain radar (MRR), is designed to measure rainfall rather than snowfall using the backscattering and vertical velocity information. Yet, the raw Doppler spectra can be reprocessed using Maahn and Kollias [2012]'s improved and innovative processing chain for data collected in snow to retrieve Doppler radar moments such as

reflectivity Z and Doppler velocity. Once mapped to X-band reflectivity this can be converted to snowfall rate S by means of a Z-S power law fitted to the local conditions using the weighing gauge information or parameterizations from existing literature (for more details see Grazioli et al. [2017a]). The 2$^{nd}$ instrument, a Mobile X-band Polarimetric radar (MXPol), for which extensive experience with the measurement of snow is available (Schneebeli et al. [2013], Scipión et al. [2013], Grazioli et al. [2015]), provided more detailed information and served as a control and reference for the calibration of the

method to use the MRR data. While the X-band radar could only be deployed during the summer campaign and had to be shipped back after completion in February 2016, the K-band radar could remain on site after the summer campaign, sheltered by a radome from the ferocious winter winds.

### 2.2.2 Disdrometer and MASC

The Biral VPF 730 disdrometer is also a non-capture instrument, which estimates the size and speed of airborne particles

from the diffusion and diffraction of an infrared light beam within a 400 cm$^3$ air volume. The volumetric sampling of the VPF730 presents an advantage over 2-D sampling instruments, which is that it does not miss particles having a much larger horizontal (due to strong wind) than vertical (falling) speed. The downside is that the instrument does not straightforwardly distinguish between falling and blowing snow [Bellot et al., 2011]. A Biral proprietary algorithm directly provides precipitation rates from the size – speed matrix. Because this is based on various assumption, including on the phase, shape





and density of the particles, particularly unwarranted in the atypical Antarctic environment, the database described here presents the matrices rather than the estimated precipitation.

A MASC was deployed next to the disdrometer. This instrument collects high-resolution stereoscopic photographs of snowflakes in free fall, while they cross the sampling area [Garrett et al., 2012], thus providing information about snowfall

microphysics and particle fall velocity. The MASC uses 3 identical 2448 x 2048 pixels cameras (with common focal point) with apertures and exposure times adjusted to trade off between the contrast on snowflakes photographs and motion blur effects. The resolution is about 33 μm per pixel. The cameras are triggered when a falling particle crosses two series of near-infrared sensors. A detailed description of the system and its calibration can be found in Garrett et al. [2012]. Information from disdrometers [Souverijns et al., [2014] and more particularly from MASC images, after image processing, provide

characterizations and classification of snow particles [Praz et al., 2017], that can be used to better process radar data.

### 2.2.3 Precipitation gauge, meteorology, and setting of the instruments

What fraction of snowfall a traditional precipitation gauge captures is unwarranted. On the other hand, unlike remote sensing instrument, the mass quantification of any captured snow is direct and straightforward. An OTT Pluvio2 precipitation gauge was deployed for the duration of the summer campaign. Snow falling in the instrument is definitely captured and weighted.

The instrument used here was equipped with a manufacturer-design wind shield meant to limit wind impacts on capture efficiency. Further, the instrument was relatively shielded from the strongest wind due to its location, on the roof of a container but on the side of a building. The MASC and disdrometer were deployed at the same partially sheltered site, the local meteorology of which was sampled with a local weather station. The radars were closely located, within at most 200 m meters of the other instruments. A composite picture of the various instrument and instrument setting is provided by Figure 2

of Grazioli et al. [2017a].

### 3 Data samples, data availability and conclusions

Grazioli et al. [2017a] extensively process and discuss the data from the different instruments. Further analyzes and presentation is beyond the scope of this data paper, and only a few snapshots are provided to illustrate the content of the database. Figure 1 shows the cumulative precipitation during the intensive summer campaign, as yielded by the Pluvio2

snow gauge and the processed MRR at the lowest useful level and at 741 m above sea-level. Only 28 out of 31 MRR levels are provided in the database. This is because the lowest levels below 250 m are too close to the surface and are affected by ground clutter [Maahn and Kollias, 2012] and data from the upper-most level are noisy. The processed MRR precipitation data are obtained as described in Grazioli et al. [2017a]. Censoring the Pluvio2 data for wind-induced biases such as vibrations and turbulence effects by cross referencing with the MRR data removes up to 30% of the quantities [Grazioli et



al., 2017a]. As the Pluvio2 is a standard instrument but there is no standard correction method, others might want to test

other approaches and the primary rather than the censored data are shown here and distributed in the database.

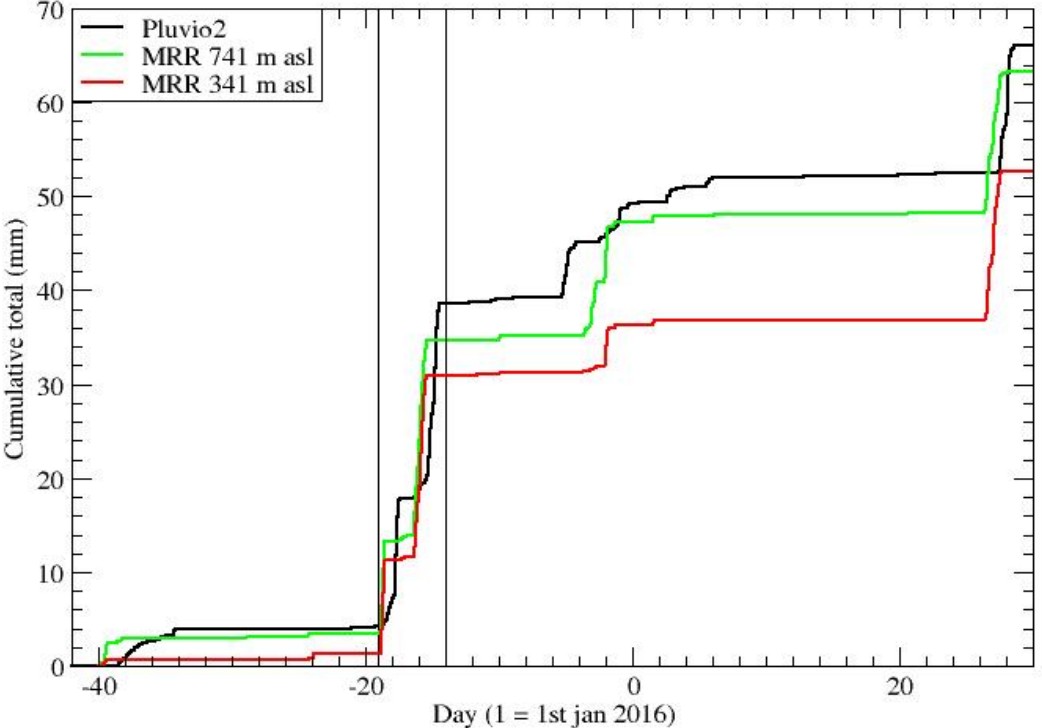

**Figure 1:** Cumulative precipitation during the APRES3 summer campaign, from the Pluvio2 and MRR instruments. Thin

black vertical lines bracket the largest precipitation event in the period, from Dec. 12 to Dec. 17 2015. Precipitation from the

MRR is reported for 2 levels above sea-level, 341 and 741 m.

The MRR precipitation at the lowest level (341 m asl) is significantly less than that at 741 m asl, showing that a significant

fraction of the precipitation formed above sublimates in the dry katabatic air layer near the surface. Further observations

show that this frequently occurs at all season of the year (see below). Meteorological and climate models suggest that at the

full scale of the Antarctic ice sheet up to 17% of the precipitation evaporates in a dry surface layer before reaching the

surface, and thus does not contribute to feed the ice sheet [Grazioli et al., 2017b]. Altogether, the 2015-16 summer was

relatively dry and few strong precipitation events occurred. One such event happened from December 12 to December 17

2015 (delineated by thin vertical black lines on Figure 1), during which the largest part of the total cumulative precipitation

this summer was recorded. Figure 2 shows an example of the Biral disdrometer size-speed matrix during this event. The

local wind was relatively strong (5.4 ms$^{-1}$ averaged on the same 10 min. as the matrix Figure 2, with significant gusts in the





period). Considering that the anemometer is set at a relatively sheltered place and thus underestimates the large-scale wind, a contribution of blowing snow to the disdrometer report is likely. However, a significant fraction of the density number of particles detected is associated with moderate speed below 4 ms$^{-1}$. Large particles (0.8 – 1.2 mm) are detected, the fall speed of which may indeed be over 1 ms$^{-1}$ as reported by the instrument.


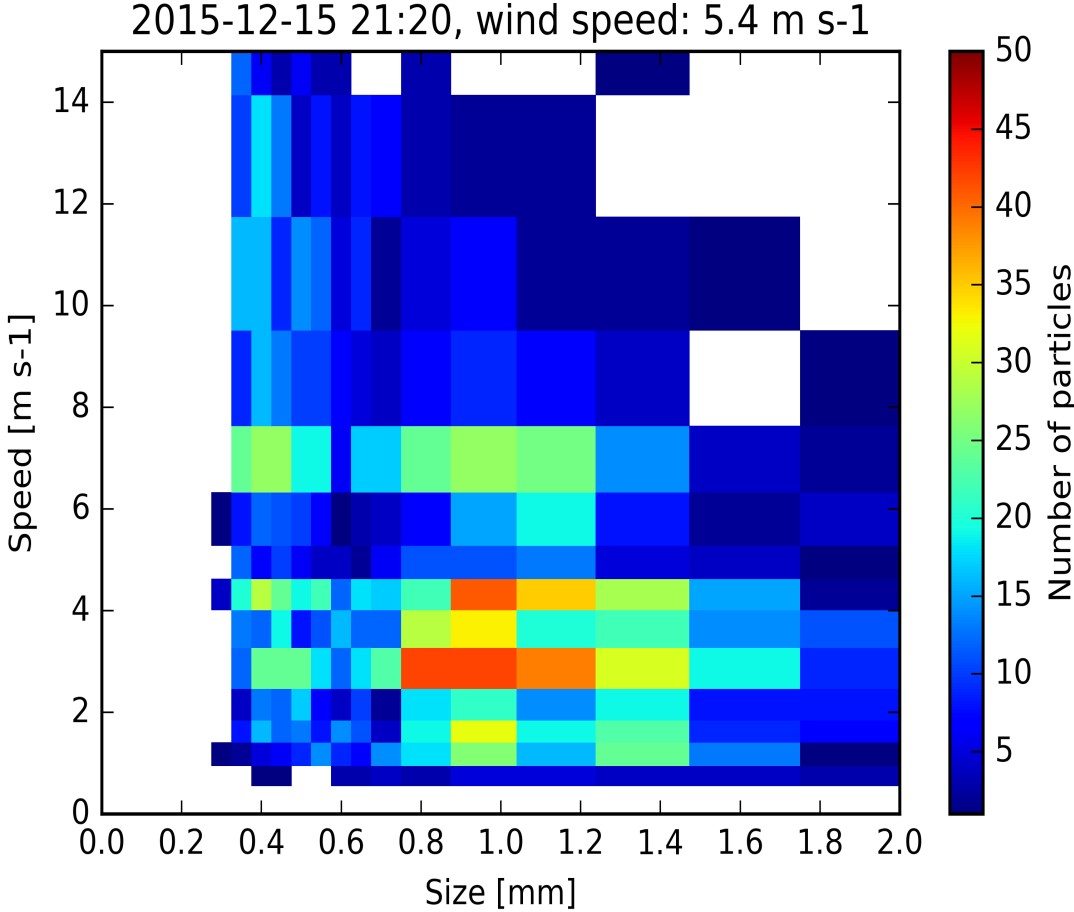

**Figure 2:** An example of the 21 x 16 fixed-levels size-speed matrix of particle density distribution from the Biral disdrometer during the large precipitation event shown Figure 1. The date and time (Dec. 15 21:20 local time) and local wind speed (5.4 ms$^{-1}$) are printed at the top of the graph.

Figure 3 shows the mean distribution (PDF) of the degree of riming of the snowfall particles as obtained by processing the MASC photographs. No less than 426229 photographs of falling snow particles were collected during the season. Each picture is processed as described in Praz et al. [2017]. The database offers the processed results in the form of a classification, rather than the photographs themselves. Figure 3 cumulates all single estimates of the degree of riming in the





database. The degree of riming is defined in this context as a continuous index between 0 (no riming on the particle detected)
and 1 (fully rimed, graupel-like particle). Almost half of the particles are close to fully rimed, indicating that cloud liquid
water is very frequent in summer. Finally, Figure 4 shows precipitation from the MRR dataset from Nov 21 2015 to Dec. 11
2016. Again, reports from 2 elevations, 341 and 741 m asl are displayed. This shows that at DDU, cumulated over a full year,
~25% of the precipitation formed in the atmosphere sublimates before reaching the surface.


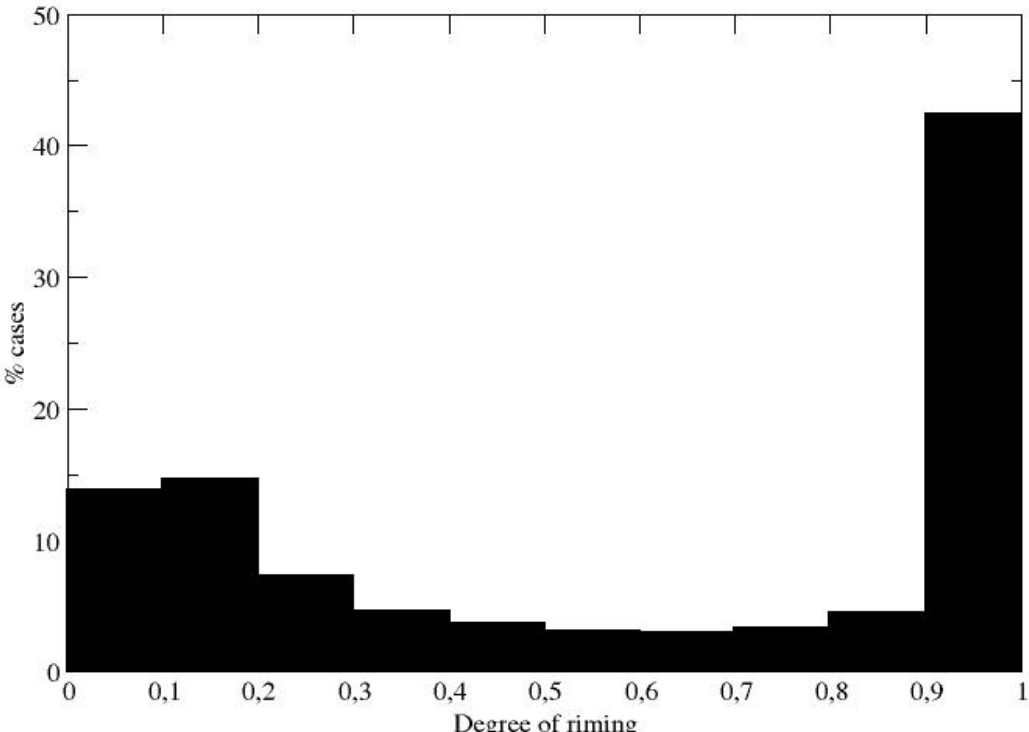

**Figure 3:** PDF of snow particle riming from the MASC data.





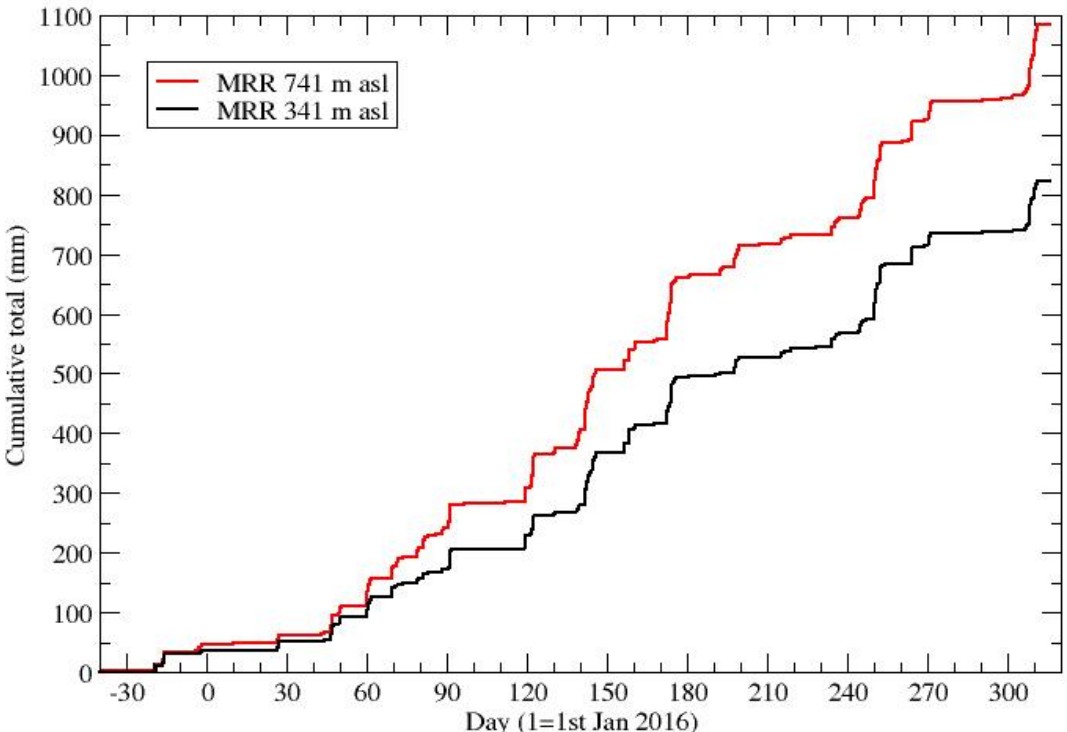

**Figure 4:** One year (Nov. 2015 – Nov 2016) of cumulative precipitation from MRR backscattering at 341 and 741 m above the surface.

| Instrument | Variables | Format | Period |
|---|---|---|---|
| Weather station | Temperature, moisture, wind | ASCII | 21-11-2015 to 06-02-2016 |
| K-band MRR radar | Precipitation profiles (28 levels) | NetCDF | 21-11-2015 to 11-11-2016 |
| Pluvio2 weighing gauge | Surface precipitation | ASCII | 17-11-2015 to 21-01-2016 |
| Biral VPF730 disdrometer | Size / speed matrices | ASCII | 02-12-2015 to 23-02-2016 |
| MASC | Snow particle classification and microphysics | ASCII | 11-11-2015 to 21-01-2016 |

**Table 1:** Summary of data from the APRES3 program available from the PANGAEA repository, Berne et al. [2017].

In conclusion, observations at DDU carried out as part of the APRS3 project provide an unprecedented dataset of



precipitation at the coast of Antarctica, complementing existing documentation efforts [Gorodeteskaya et al., 2015] in a
region which otherwise suffers a severe shortage of such data. Our analysis of the data yields new insights into the
characteristics and particularities of Antarctic snowfall, in particular that a large fraction of the precipitation formed in the
atmosphere sublimates before reaching the surface. This information could only be obtained with instruments that can profile
through the atmospheric layers, like radars here. However, the dataset goes beyond radar data and provides extensive
complementary characterization of snow particles geometry and cumulative quantities of snowfall at the surface. Except for
the dataset from the dual polarization scanning radar MXPol during the summer campaign, the size of which (about 4TB) is
too large to be shared on-line but can be obtained by direct request to the authors, all data are now distributed [Berne et al.
2017] and can be freely accessed from PANGAEA repository (https://www.pangaea.de, DOI: 10.1594/PANGAEA.883562).
Table 1 provides a summary of the the variables and periods covered and distributed on line for each instrument. At time of
writing this paper, the project carries on with continuous collection of precipitation profiles with the MRR, and a planned
contribution to the Year Of Polar Prediction (YOPP, http://www.polarprediction.net/) international austral special coordinated
observation period from Nov 2018 to Feb 2019, the data from which will also be made available to the community. It may be
noticed that because of a significant weather service (Météo France) involvement including additional radiosoundings, in
addition to the planned APRES3 contribution, DDU is identified as one of the YOPP observation hot spots for the special
observing period.


**Data availability:** The APRES3 database is available in open access on PANGAEA,  DOI: 10.1594/PANGAEA.883562

**Acknowledgements:** We thank the French Polar Institute, which logistically supports the APRES3 measurement campaigns.
We also acknowledge the support of the French National Research Agency (ANR) to the APRES3 project. The project
Expecting Earth-Care, Learning fro**Data availability:** m the ATrain (EECLAT) funded by the Centre National d'Etudes
Spatiales (CNES) also contributed support to this work. The Swiss National Science foundation SNF is acknowledged for
grant 200021_163287, financing the Swiss participation to the project. PANGAEA is gratefully acknowledged for hosting
and distributing the APRES3 data.

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
