# Peer review of "Precipitation at Dumont d'Urville, Adélie Land, East Antarctica: the APRES3 dataset"

_Earth System Science Data, 2018_

## Referee Comment (RC1) · S. Colwell (Referee) · 17 May 2018

Please remove the word "the" from before Antarctic precipitation on the 3rd line of the abstract.

———————————————

---

## Referee Comment (RC2) · Anonymous Referee #2 · 22 May 2018

The authors describe a precipitation data set obtained during the APRES3 campaign in Antarctica. Given the lack of precipitation measurement in Antarctica, the data set is highly relevant and I strongly appreciate that the authors share this unique data set. In general, I recommend the paper for publication subject to the following minor comments.

General: The files provided to not have a common format. With libraries such as xarray and pandas it is actually very easy to provide consistent CSV files or even netCDFs. I would appreciate if the authors would provide at least reading routines.

Errors: I understand that a precise uncertainty estimation is not possible for every variable, but at least add (or point to) a discussion of uncertainties for every data set.

[Figure]

MASC data set: unit of fall speed is missing

L12: is -> was (?)

L 26f: 'poor cousin', L 43: 'can simply not done', L 74 "run into problems": too colloquial

L 45f: I would recommend: "However, strong katabatic winds are frequently blowing at the peripheries."

L 50: re-mobilized

L 103: masterpieces -> the core instruments

L 101: Please add information about the scan strategy of the X-band radar. What range resolution was used for the MRR and the X-band?

L 106: Add that it is a pulsed radar in contrast to the FMCW.

L 117: What is the attenuation of the radome?

L 151: The reason why the lowest two bins have to be removed is not related to ground clutter, they are also too noisy.

L 152: I would recommend: "Precipitation rates were retrieved from MRR data following Grazioli et al". Further, I would recommend to summarize the retrieval technique in one sentence.

L155: 'standard correction method' for what? Wind?

L255: Remove 'data availability'
* * *

---

## Referee Comment (RC3) · Anonymous Referee #3 · 2 Jun 2018

The paper describes a rare data set describing Antarctic precipitation with the help of different in-situ and remote sensing measurements. The data are available for download at Pangea and will certainly be of high interest for the community. However, the description of the data (available for download at Pangea) needs clarification and should be more precise and better structured before the paper can be published.

MAJOR COMMENTS

In the beginning, it is unclear what the data set really consists of. Is it about a field campaign which I understood to be a subset of APRES3 (program also including satellite data) but then the title of the paper is not correct. Is the X-Band data part of the data set? If yes, what was its actual operation time. A time line with the different instruments should be given which also provides information on data availability, e.g. start and stop,

any interruptions? For clarification Table 1 should be moved to the beginning and either be expanded or another table added to provide more details on data availability, resolution and uncertainty.

In general, I am missing a clear summary about the data, e.g. resolution (temporal, spatial, spectral), scanning routine, uncertainty, level of processing/quality control. A section on available output and data processing might be appropriate. I am still left with questions like the noise level of the MRR, the exact binning and the exact conversion of K-band radar reflectivities to precipitation? This is important as no reflectivity data are provided. Why not?

It was not that easy to downloaded the different data streams and I could not find all necessary information, e.g. is time stamp the beginning or end of the measurements? Why is the time array for MRR so complicated?

MINOR COMMENTS

Who organized, financed (acknowledgement?) and performed APRES3?

Abstract should also clearly say which data are presented.

L27: " Antarctica is the poor cousin of global precipitation observation and climatology building efforts." Not really true as Antarctica is part of the globe, also other desert areas are challenging...I recommend a simple statement on the challenges.

L37: not only in situ but also ground-based remote sensing (as you will demonstrate later) is needed. Maybe you can cite Maahn et al., JGR, 2014 about the Cloudsat blind zone to substantiate the statement.

L45-L50: a reference is needed about gauge problems for snowfall

L67: When did the intensive observation campaign took place? What was different there? Timeline?

L76: What are the data?

L77: "Further work is ongoing to address the calibration, verification and validation of meteorological and climate models and of satellite remote sensing techniques with the data." What is the difference between verification and validation? Maybe say which part is related to measurements, retrievals, model, satellites?

L87: Altitude above mean sea level?

L106: Say that MRR is vertically staring

L110: Say that most Ze-S relations for radars have been derived for 10, 35 GHz or 94 GHz and therefore the measured equivalent radar reflectivity at 24 GHz is first converted to (how?). There is little information on the uncertainty of the derived product - I know it is difficult but people need to be aware.

L124: resolution of the matrix?

L134: Souverijns et al 2017 not 2014

L137: The subsection 2.2.3 has meteorology in the title but nothing is said about it? As the data are part of the described data set a short summary (table) should be provided.

L139: What does definitely mean? - nothing can be sublimated?

L142: Is the container lower than the building? Though a figure is shown in a different publication a sketch or a reproduction of the figure would be helpful to better understand the setup.

L 146: The structure of this section is not obvious. Details on MRR range gates are given together with examples but why is the precipitation time series shown at the beginning and at the end?

L150 : The numbering of the provided MRR bins and range resolution are not clear. How are the vertical ranges exactly defined? Why is height 741 chosen? Why are just the upper two? bins so noisy? The minimum detectability should only slightly increase with altitude. What is the minimum detectable

reflectivity anyhow?  Maybe provide 2D histogram as Kneifel et al., 2011 https://link.springer.com/content/pdf/10.1007%2Fs00703-011-0142-z.pdf.

L155: why not explain: "..2017a] as visible by the accumulation of the Pluvio in the time periods between snowfall events.

L166: From Figure 1 one could derive also 17 % as the 60 vs 52 mm are accumulated over this time.

L326: Make clear that this is no article or letter-

Figures: The quality is in general poor - a different graphics format might help.

Figure 3: Which time period?

Figure 4: Why is the Pluvio not shown? Why not show this overview figure first and make an inlet for the short period shown in Fig. 1?

LANGUAGE AND GRAMMAR

Abstract: " a result evidenced thanks to the profiling capabilities of precipitation radars" -> as derived from profiling radar measurements.

L26: "to verify the current precipitation in the models are still in demand" I don't think verification is possible.. -> to raise confidence in precipitation simulations.

L 39: water equivalent

L42: genial

L54: for and An ...

L66: Hypothesis should be substantiated by calibration and validation

L94: "..including THE relation with precipitation" L98: METAR acronym

L102: radars are key ingredients of APRES

L180: spell out acronym pdf - by definition it is a "mean"

L217: skip the first part. "Because a significant.."

L225: Data availability in the middle of the sentence

---

## Author Comment (AC1) · 23 Jul 2018

The authors wish to thank all 3 referees for their comments, pointing for weaknesses, and suggested improvements, which contributed to significantly improve the paper.

Our responses and description of how we account for the comments is listed in the attached pdf file.

Please also note the supplement to this comment: https://www.earth-syst-sci-data-discuss.net/essd-2018-34/essd-2018-34-AC1-supplement.pdf

---

## Author Comment (AC2) · 23 Jul 2018

**Accounting of reviewer's comments**

Reviewer comment / request in blue

Report on our accounting of referee's comments / requests in red

**Referee 1 :** S. Colwell

Please remove the word "the" from before Antarctic precipitation on the 3rd line of the abstract.

OK done

Thank you Steve

**Anonymous Referee 2:**

The authors describe a precipitation data set obtained during the APRES3 campaign in Antarctica. Given the lack of precipitation measurement in Antarctica, the data set is highly relevant and I strongly appreciate that the authors share this unique data set. In general, I recommend the paper for publication subject to the following minor comments.

General: The files provided to not have a common format. With libraries such as xarray and pandas it is actually very easy to provide consistent CSV files or even netCDFs. I would appreciate if the authors would provide at least reading routines.

We now provide all datasets on Pangea both in "native" and (if native is not cdf) in netcdf format. We think the basic (ascii) format is useful for those who do not practice python or other (e.g. ferret) tools that can read and process netcdf or specialized format. This is now reported in the text.

Errors: I understand that a precise uncertainty estimation is not possible for every variable, but at least add (or point to) a discussion of uncertainties for every data set.

Unfortunately, uncertainties are hard to estimate, particularly in the special environment we work in. Except for the meteorological instruments on the weather station, the notice provided with the commercial instruments used in this study (radars, MASC, disdrometer) do not provide any uncertainty numbers. Mostly, these are "black boxes" and the user has little handle to estimate uncertainty. The MASC processing is based on expert system, which does not provide uncertainties (see Praz et al. [2018]). Some uncertainies are quoted in Grazioli et al. ([2017a], their section 2.2.1) but this is admittedly limited.

A note on uncertainties, and limited estimation of, is now provided in the paper. Concerning the Vaisala weather transmitter (see below), the reader is referred to the manufacturer's notice.

MASC data set: unit of fall speed is missing

OK, this is corrected

L12: is -> was (?)

No. The program is still operational, until the end of 2019. The main observation component is now limited to one MRR but there are still snapshot deployments of other instruments Additional data will also be distributed at some point, in particular those obtained as part of a contribution of APRES3 to the southern Year Of Polar Prediction (YOPP) SOP. There is also work to exploit the data to improve modeling and remote sensing of antarctic precipitation, which is also part of APRES3 but not of the observation component of APRES3.

L 26f: 'poor cousin', L 43: 'can simply not done', L 74 "run into problems": too colloquial

"can simply not be done" is replaced by "is not possible"
"run into problems" is replaced by "had problems"
However, we keep the "poor cousin" which we think best conveys not only the fact that Antarctica is undersampled compared to other continents, but also that the undersampling is partially because of undercomitment compared to the other regions. The other referees do not complain, and referee 3 even picks up the expression to discuss an other aspect of the paper.

L 45f: I would recommend: "However, strong katabatic winds are frequently blowing at the peripheries."

OK done

L 50: re-mobilized

OK, done

L 103: masterpieces -> the core instruments

OK done

L 101: Please add information about the scan strategy of the X-band radar. What range resolution was used for the MRR and the X-band?

This is already described in Grazioli et al. (2017a, their section 2.2). A summary of essential information is now provided in the paper: 75 m radial resolution, maximum radial distance : 30 km + different types of scans within a repeating scanning sequence of 5 min: Plan Position Indicator (PPI), Range Heigh Indicator (RHI) + vertical profiles.

L 106: Add that it is a pulsed radar in contrast to the FMCW.

OK done

L 117: What is the attenuation of the radome?

Figure 4 of Grazioli et al. [2017a] shows a scatter plot of reflectivity Z from MX-Pol and MRR. The slope is very close to 1 (0.99) and the intercept is 6.14 dBZ. This is the radome attenuation, used to map the 24.3 GHz MRR reflectivity to the 9.41 GHz MXPol reflectivity in equation (1). The radome attenuation is now reported in the paper.

L 151: The reason why the lowest two bins have to be removed is not related to ground clutter, they are also too noisy.

The referee is right stating that removing the 2 lowest bins is not related to ground clutter. However the word "noise" is maybe not so relevant either. In fact, following Peters et al. [2005], data processing is based on assumptions that are not valid for the two first gates and it may lead to overestimation of reflectivity.This is now corrected in the text, Peters et al. [2005] is added in the reference list.

Peters, G., B. Fischer, H. Münster , M. Clemens, A. Wagner, 2005. Profiles of raindrop size distributions as retrieved by Microrain Radars. *J. Appl. Meteorol.* **44**, 1930–1949, doi: 10.1175/JAM2316.1

L 152: I would recommend: "Precipitation rates were retrieved from MRR data following Grazioli et al". Further, I would recommend to summarize the retrieval technique in one sentence.

Retrieval method is now summarized in one sentence: ...following Grazioli et al [2017]: the reflectivity was converted into liquid water equivalent rate by fitting the prefactor and exponent of a Z-S relationship using carefully filtered nearby weighing gauge (Pluvio2) data.

L155: 'standard correction method' for what? Wind?

Yes, for under or overcatch due to wind. "for wind effect" added in the text.

L255: Remove 'data availability'

OK done

Thank you for a thoughtful review that definitely helped further improve the paper.

**Anonymous Referee #3**

The paper describes a rare data set describing Antarctic precipitation with the help of different in situ and remote sensing measurements. The data are available for download at Pangea and will certainly be of high interest for the community. However, the description of the data (available for download at Pangea) needs clarification and should be more precise and better structured before the paper can be published.

MAJOR COMMENTS
In the beginning, it is unclear what the data set really consists of. Is it about a field campaign which I understood to be a subset of APRES3 (program also including satellite data) but then the title of the paper is not correct.

That is correct, the APRES3 program also includes satellite remote sensing and modeling activities. The dataset is about observations made from the surface in the framework of the program, by the APRES3 team, which unlike the satellite data are not available from other sources. The title is modified to "Precipitation at Dumont d'Urville, Adélie Land, East Antarctica: the APRES3 field campaigns dataset".

 Is the X-Band data part of the data set? If yes, what was its actual operation time. A time line with the different instruments should be given which also provides information on data availability, e.g. start and stop, any interruptions? For clarification Table 1 should be moved to the beginning and either be expanded or another table added to provide more details on data availability, resolution and uncertainty.

Yes, the X-Band data is part of the dataset but is too big to be conveniently hosted and distributed at Pangaea. This was (and still is) mentioned in section 4 and in the Data Availability section: "Except for the dataset from the dual polarization scanning radar MXPol during the summer campaign, the size of which (about 4TB) is too large to be shared on-line but can be obtained by direct request to the authors..."

Table 1 does provide time line and availability (which is basically the same, start and end dates are provided in the last column). We now add columns with time and space resolution. Concerning uncertainty, please see our response to referee 2, who specifically raises the point, and the added information in the text. Not being able to provide more on uncertainties is certainly frustrating, but we do not think of a way for better assessment.

We agree that Table 1 should show earlier in the text and we move it to the beginning of section 3. On the other hand, it does not seem appropriate to have it at the very beginning of the paper before we describe and discuss the nature and instrumental aspects of the data in section 2.2.

In general, I am missing a clear summary about the data, e.g. resolution (temporal, spatial, spectral), scanning routine, uncertainty, level of processing/quality control. A section on available output and data processing might be appropriate. I am still left with questions like the noise level of the MRR, the exact binning and the exact conversion of K-band radar reflectivities to precipitation? This is important as no reflectivity data are provided. Why not?

Much of this information is not reproduced in this data paper as it would essentially duplicate Grazioli et al. [2017a] (with some risk of "plagiarism" bell ringing in the editorial room...). In response to this and the previous referee, we do provide more information in the paper, such as the summary sentence "..following Grazioli et al (2017): the reflectivity was converted into liquid water equivalent rate by fitting the prefactor and exponent of a Z-S relationship using carefully filtered nearby weighing gauge (Pluvio2) data." and summary of essential information for MXPol: sampling strategy "75 m radial resolution, maximum radial distance : 30 km + different types of scans within a repeating scanning sequence of 5 min: Plan Position Indicator (PPI), Range Height Indicator (RHI) + vertical profiles".

The raw (reflectivity, doppler velocity) data are not provided in line because of the size of the dataset. This may admittedly be of interest to radar data processing experts. We do not advertise the raw data as part of the data set (table 1) but now mention in the text that it can be made available by request to the authors.

It was not that easy to downloaded the different data streams and I could not find all necessary information, e.g. is time stamp the beginning or end of the measurements?

We are not sure we understand this remark. Time stamps are provided for all files, e.g. time stamp for beginning and end are provided on the 1[st] and last line of all ascii files, and time arrays are provided along with the data arrays in the NetCDF file.

 Why is the time array for MRR so complicated

There are 3 variables related to time: (1) time in Julian format, (2) time in seconds since 1970, (3) time resolution of the measurements (always 1h for the local relation). (1) and (2) are provided although they are redundant, to provide 2 commonly used numerical standards.

MINOR COMMENTS

Who organized, financed (acknowledgement?) and performed APRES3? Abstract should also clearly say which data are presented.

We have an acknowledgment section, located at the end of the paper as usual: "We thank the French Polar Institute, which logistically supports the APRES3 measurement campaigns. We particularly acknowledge the support of the French National Research Agency (ANR) to the APRES3 project. The project Expecting Earth-Care, Learning from the ATrain (EECLAT) funded by the Centre National d'Etudes Spatiales (CNES) also contributed support to this work. The Swiss National Science foundation SNF is acknowledged for grant 200021_163287, financing the Swiss participation to the project. PANGAEA is gratefully acknowledged for hosting and distributing the APRES3 data".

The abstract now provides a list of types of instruments used.

L27: " Antarctica is the poor cousin of global precipitation observation and climatology building efforts." Not really true as Antarctica is part of the globe, also other desert areas are challenging...I recommend a simple statement on the challenges.

Even though the observation of precipitation is also sparse over other lands, it is particularly critical in Antarctica at the continental scale. Antarctica is the only continent for which the GPCC refrains from providing a precipitation analysis based on in situ reports, blaming insufficient data. This is the only conterminous land surface left aside in GPCC's otherwise global analysis. There is thus a special case made for Antarctica. We slightly rephrase this part of the paper to make it clearer that the "poor cousin" reflects GPCC's assesment.

L37: not only in situ but also ground-based remote sensing (as you will demonstrate later) is needed. Maybe you can cite Maahn et al., JGR, 2014 about the Cloudsat blind zone to substantiate the statement.

This remark does not seem to fit with L37. May be line 34? In that case, this is the very beginning of the paper, we have not yet introduced the concept of surface remote sensing and all related issues, referring to the blind zone at this point would be premature. We replace "in situ" by "ground based" which implicitly involves "in situ" but again, it is too early in the paper to refer to "ground based remote sensing"

L45-L50: a reference is needed about gauge problems for snowfall

This is provided in the introductory (first) sentence of the paragraph: The measurement of solid precipitation is notoriously difficult (Goodison et al. [1998], Nilu [2013])

L67: When did the intensive observation campaign took place? What was different there? Timeline?

Again, we are not sure we understand the comment. L67 precisely reports the timing of the intensive observation period: "starting in November 2015 until February 2016 ». The dataset description starts just a few lines below (L82).

L76: What are the data?

Data description starts just 6 lines below.

L77: "Further work is ongoing to address the calibration, verification and validation of meteorological and climate models and of satellite remote sensing techniques with the data." What

is the difference between verification and validation? Maybe say which part is related to measurements, retrievals, model, satellites?

OK "verification" removed, "(snow fall occurrence and rates, vertical profiles)" added.

L87: Altitude above mean sea level?

41 m on average, this is now reported in the paper.

L106: Say that MRR is vertically staring

OK, done

L110: Say that most Ze-S relations for radars have been derived for 10, 35 GHz or 94 GHz and therefore the measured equivalent radar reflectivity at 24 GHz is first converted to (how?).

OK done. With respect to "how" (details), we refer to Grazioli et al. [2017] and do not replicate all thorough information already available therein.

There is little information on the uncertainty of the derived product - I know it is difficult but people need to be aware.

Again, see response to referee 2 above and information added in the text.

L124: resolution of the matrix?

The resolution is not regular and can be figured out from figure 2.

L134: Souverijns et al 2017 not 2014

OK corrected

L137: The subsection 2.2.3 has meteorology in the title but nothing is said about it? As the data are part of the described data set a short summary (table) should be provided.

This remark more probably refers to L143 rather than L37. We now provide information on local meterology sampling: "the local meteorology of which was sampled locally by a Vaisala WXT520 weather transmitter, principles, instrumental accuracy and performance of which can be found in the manufacturer's User's Guide (https://www.vaisala.com/sites/default/files/documents/M210906EN-C.pdf). Note that this station integrates an acoustic rain gauge, not appropriate to measure snow fall, thus the deployment of the Pluvio2". As this is an integrated weather station, no table listing of individual instrument is provided.

L139: What does definitely mean? - nothing can be sublimated?

No, nothing can be sublimated. The gauge is filled with antifreeze fluid which prevents any sublimation after capture

L142: Is the container lower than the building? Though a figure is shown in a different publication a sketch or a reproduction of the figure would be helpful to better understand the setup.

Yes, the container is slightly lower than the building. A figure is added (new Figure 1, shifting other figure numbering by 1) which shows the general setting

L 146: The structure of this section is not obvious. Details on MRR range gates are given together with examples but why is the precipitation time series shown at the beginning and at the end?

The time series is 1$^{st}$ shown at the beginning, limited the summer intensive observation period. This is the period for which the dataset we distribute includes a whole range of instruments – illustrated Figure 2 comparing MRR and Pluvio2. Then, over a longer period of time (a full year), only the MRR operates – thus Figure 5 only showing MRR results, but over a full year.

L150 : The numbering of the provided MRR bins and range resolution are not clear. How are the vertical ranges exactly defined? Why is height 741 chosen?

We are not sure we understand the issue made about numbering. The vertical resolution (100 m) for the MRR is now first provided in section 2.2.1, along with similar information for the MXPol radar. As we now provide the site elevation above sea-level (as requested by the referee), the "41" that enters the gates altitude probably makes more sense The height 741 m is selected as the closest to the mean height of maximum snowfall before evaporation occurs, as reported in Grazioli et al. ([2017b], their Figure 2).

Why are just the upper two? bins so noisy? The minimum detectability should only slightly increase with altitude. What is the minimum detectable reflectivity anyhow?

Actually, only the upper level is considered noisy. According to Peters et al. [2005] (last line of their Table 2): "Min detectable radar reflectivity [here for z = 1000 m, 100 m intervals, DelatT= 60 s] : -2 dBZ". Also Kneifel et al. [2011] report that : "The detectability is highest close to the ground with -2 dBz (35 GHz equivalent) at 500 m, but is decreasing with height to 3 dBz at 3,000 m".  This is now reported in the paper.

Maybe provide 2D histogram as Kneifel et al., 2011
https://link.springer.com/content/pdf/10.1007%2Fs00703-011-0142-z.pdf.

Thank you for the suggestion but we are not sure which 2D histogram in Kneifel et al. The only reference to a histogram in Kneifel et al. is their Figure 5 but this is not 2D. Other figures that may qualify as 2D histograms are height – Ze graphics but we do not show any Ze, only processed data.

L155: why not explain: "..2017a] as visible by the accumulation of the Pluvio in the time periods between snowfall events.

OK done

L166: From Figure 1 one could derive also 17 % as the 60 vs 52 mm are accumulated over this time.

Right, but here we discuss annual and full continental scales

L326: Make clear that this is no article or letter-

Right, "News in focus" now reported. However, it will be up to the editor to decide exactly how this must be referenced.

Figures: The quality is in general poor - a different graphics format might help.

Right, figures of limited resolution are provided with the submitted and revised version to limit file size but full resolution will be provided at a final stage if the paper is accepted for publication.

Figure 3: Which time period?

This is over the full time period the instrument was operating. This was mentioned in the main text (" Figure 3 cumulates all single estimates of the degree of riming in the database")  and is now also reported in the legend.

Figure 4: Why is the Pluvio not shown? Why not show this overview figure first and make an inlet for the short period shown in Fig. 1?

This is because the Pluvio2 only operated during the intensive observation period and was removed at the end of Jan 2016, see Table 1.

LANGUAGE AND GRAMMAR

Abstract: " a result evidenced thanks to the profiling capabilities of precipitation radars" -> as derived from profiling radar measurements.

OK

L26: "to verify the current precipitation in the models are still in demand" I don't think verification is possible.. -> to raise confidence in precipitation simulations.

OK

L 39: water equivalent

OK

L42: genial

Changed to "hospitable"

L54: for and An …

OK

L66: Hypothesis should be substantiated by calibration and validation

This comment does not seem to fit with L66. Should it be L64? In which case, OK done.

L94: "..including THE relation with precipitation"

Here again, comment does not fit with L94.

L98: METAR acronym

METeorological Aviation Reports, now explicated in the text

L102: radars are key ingredients of APRES

"are the core instruments" following suggestion by referee 2

L180: spell out acronym pdf - by definition it is a "mean"

OK done

L217: skip the first part. "Because a significant.."

OK done

L225: Data availability in the middle of the sentence

OK, removed.

Thank you for a thoughtful review that definitely helped further improve the paper.